# Seaweed Potential in the Animal Feed: A Review

**Tiago Morais [1], Ana Inácio [2,†], Tiago Coutinho [2,†], Mariana Ministro [3], João Cotas [4], Leonel Pereira [4] and Kiril Bahcevandziev [2,\*]**

[1] Lusalgae, Lda, Incubadora de Empresas da Figueira da Foz, Rua das Acácias N° 40-A, 3090-380 Figueira da Foz, Portugal; tsmorais@lusalgae.pt

[2] Agricultural College of Coimbra (ESAC/IPC), Research Centre for Natural Resources Environment and Society (CERNAS), Institute of Applied Research (IIA), 3045-601 Coimbra, Portugal.; anacmi.31@gmail.com (A.I.); tiago28coutinho@hotmail.com (T.C.)

[3] Polytechnic Institute of Coimbra/ISEC, Rua Pedro Nunes, Quinta da Nora, 3030-199 Coimbra, Portugal; marianaministro95@gmail.com

[4] Department of Life Sciences, MARE—Marine and Environmental Sciences Centre, University of Coimbra, 3001-456 Coimbra, Portugal; jcotas@gmail.com (J.C.); leonel.pereira@uc.pt (L.P.)

\* Correspondence: kiril@esac.pt

† Co-second authors.

**Abstract:** Seaweed (known as marine algae) has a tradition of being part of the animal feed in the coastal areas, from ancient times. Seaweeds, are mixed with animal feed, because when consumed alone can have negative impact on animals. Thus, seaweeds are very rich in useful metabolites (pigments, carotenoids, phlorotannins, polyunsaturated fatty acids, agar, alginate and carrageenan) and minerals (iodine, zinc, sodium, calcium, manganese, iron, selenium), being considered as a natural source of additives that can substitute the antibiotic usage in various animals. In this review, we describe the nutritional values of seaweeds and the seaweed effects in the seaweed-based animal feed/supplements.

**Keywords:** seaweeds; feed additive; feed supplement; animal nutrition

## 1. Introduction

Seaweeds colonize aquatic habitats and are used mainly by coastal populations [1,2]. Many seaweed species are normally used in unprocessed form, in medicine, human diets, animals feeds and for improvements in agricultural soil as fertilizers [3]. They are rich in potassium, sodium, calcium, magnesium and phosphorus and are a source of essential trace elements, such as iron, manganese, copper, zinc, cobalt, selenium and iodine [4]. Little is known about their bioavailability in nutrients [5]. Seaweeds are simple organisms, which are able to take advantage of sunlight to convert carbon dioxide into sugars and oxygen, during the photosynthesis process. The most common varieties of edible algae include: *Neopyropia/Porphyra/Pyropia* spp., *Undaria pinnatifida*, *Saccharina latissima*, *Palmaria palmata* and *Chondrus crispus*, varieties that are associated with many health benefits, such as decreasing blood pressure, preventing spills and they are a valuable protein source [6].

Seaweed biomass is a valuable alternative ingredient for livestock. Macroalgae, in general, show great differences in proteins, minerals, lipids and fibers. The high mineral content of seaweeds is due to their ability to absorb inorganic substances from the environment; they contain a small amount of lipids, mainly polyunsaturated fatty acids (PUFAs), although they are rich in polysaccharides. Seaweed has only a small percentage of lipids (1–5%), but the majority of those are PUFAs. Predominantly, brown and red seaweed contain more PUFAs 20:5 n-3 (EPA) and 20:4 n-5 (arachidonic acid) than green algae [7–9]. Seaweed has a highly variable composition, which depends on the species, time of collection, habitat and on external

conditions such as water, temperature, light intensity and nutrient concentration in water [9,10]. Algae contain high-levels of nonprotein nitrogen, such as free nitrates, resulting in nitrogen-to-protein conversion factors of 5.38, 4.92 and 5.13 for brown, red and green seaweed, respectively [8,9]. There are various edible seaweeds for human consumption with high protein content, with variable essential amino acids [11]. They also absorb minerals from seawater and contain 10 to 20 times more than the land plants [12]. In general, it is accepted that green and red algae have higher nutritional value than brown algae due to low protein and high mineral content. However, brown algae have a higher and diversified content on bioactive molecules with high commercial interest [9]. Therefore, algae can provide energy, minerals and proteins to animal feed and have potential as alternative protein source for ruminants [13].

The analysis of the protein quality and concentration is essential to determine the nutritional value of the algae biomass, so it can be used fresh, dried, liquefied or cooked. This analysis is important as it identifies the concentration of essential amino acids (EAAs) [14].

Algae have a relatively high protein quality compared to cereal and soy flour. More than 75% of seaweed has higher proportions of total essential amino acids than wheat flour and 50% higher than soy flour and also higher than rice and corn [14–16]. The proportion of EAAs methionine and lysine are comparable to traditional protein sources. Algae are generally richer than soy flour in the proportion of methionine but poorer when compared to wheat flour. On the other hand, algae have a lower proportion of lysine than soy and wheat flour [14].

The main limitation for the use of seaweed proteins is the concentration of EAAs and not the quality of the total amino acids or proteins. However, the concentration of EAAs in seaweed, in an entire biomass base, is considerably lower than in traditional sources, such as corn and soy, thus it is not suitable as a protein source in compound diets for monogastric animals. This does not detract from its positive health benefits for humans and livestock, where its few calories and high mineral content may be desirable [14].

Among the marine organisms, seaweed represents one of the richest sources of natural antioxidants and antimicrobials. They are also an excellent source of vitamins such as A, Bl, B12, C, D and E, riboflavin, niacin, pantothanic acid, folic acid as well as minerals such as Ca, P, Na, K and I [17].

In short, new alternatives to reduce or replace the use of antibiotics in animal diet is needed. Thus, we can contribute to find a natural product that does not only eliminate or prevent diseases but also improves the nutrient quality of meat and eggs. With the continued study of the sea resources, numerous species of algae with favorable biological activity have been reported as acceptable for inclusion in diets for rats, chickens, laying hens and pigs [18].

In this review, our intention is to analyze the seaweed potential for animal feed and to contribute to the development of its standardized use, reducing the animal health risks.

## 2. Seaweeds: Nutritional Profile

As demonstrated earlier, seaweeds (or macroalgae) are divided into three large groups, without any taxonomic value, based on the color they present [19,20]. There are 10,100 seaweed species known worldwide and they can be observed in all seawater habitats with some seaweeds appearing in freshwater [21].

Green seaweeds (Chlorophyta), of which there are known to be 2200 species, at maximum reach 1 m in height. Their color is related to the presence of chlorophyll [4,21,22]. Red seaweeds (Rhodophyta), with 6100 species, are efficient in photosynthesizing in deeper waters. Their length varies and they are similar to green seaweeds. Their color results from the presence of pigments, phycoerythrin and phycocyanin, which disguise β-carotene, lutein, zeaxanthin and chlorophyll [4,21,23–25]. For brown seaweeds (Ochrophyta, Phaeophyceae), with nearly 1800 species, only 1% is known to exist in fresh water and their length can be up to 50 m. The brown color is related to their content of carotenoid fucoxanthin, which disguises β-carotene, violaxanthin, diatoxanthin and chlorophyll.

The main seaweed polysaccharides are laminarin, fucoidans, agar, carrageenans, porphyran, ulvans and alginates [4,21,26,27].

Seaweed mainly contains high levels of glutamic acid, present in both free and protein-bound forms, contributing to typical flavor known as umami. They also contain various bioactive amino acids and peptides, such as taurine, carnosine and glutathione [5]. In this point we intend to address the nutritional value and other relevant molecules found in the different seaweeds.

## 2.1. Green Seaweeds

The commonly known green algae are organisms which belong to the Chlorophyceae class (phylum Chlorophyceae), including both microscopic and macroscopic species. They are the most diverse algae group, with more than 13,000 species; it is estimated that about half of these species are seaweeds. The characteristic color is due to the presence of the chlorophyll *a* and *b*, pigment used during the photosynthetic process [2,28].

Their color usually depends on the balance between these chlorophylls and other pigments, such as β-carotene and xanthophylls. Green algae are common in areas where light is abundant, such as shallow water and natural pools. The main genus include *Ulva*, *Codium*, *Chaetomorpha* and *Cladophora* [4].

*Ulva* is a one of the most common genera of green seaweeds, also found in brackish water (mainly in estuaries). Being filled with minerals, proteins and vitamins, these species are very appealing to study at a nutritional level [3]. *Ulva*'s biomass is relatively rich in proteins (Table 1) and has a potential as an alternative source of proteins for animal feeding, contains highly insoluble dietary fibers (glucans) and soluble fibers, having higher protein content than other green seaweeds [4].

*Ulva* sp. grows abundantly in areas rich in nutrients, float together along the coast, block watercourses and destroy the marine ecosystem, which becomes a serious threat to the fishing industry and the development of tourism. Consequently, it could be of great practical interest to make *Ulva* waste profitable. This seaweed is not studied only for its high protein levels. The interest, both academic as commercial, in the abundant and highly sulfated ulvans that are extracted from *Ulva*, has increased in recent years. Ulvan is a heteropolysaccharide of the cell wall that represents 9 to 36% in dry weight of *Ulva* sp. biomass [29]. Ulvan consists of rhamnose, xylose, glucose, uranic acid and sulfate, which regulate immune functions and act as an antioxidant and antibiotic [22]. A high level of this sulfated polysaccharide in *Ulva* sp. reveals its anticoagulant, antiviral, anti-inflammatory, antihyperlipidemic, immunomodulatory and anticancer activities [18].

Some *Ulva* species are used as livestock feed [30] and adding *Ulva* to diets in powder form can decrease an abdominal and subcutaneous fat, improving meat quality and amylase activity in the duodenal content of chicken [5].

The use of *Ulva* as a food or animal feed is a daunting task since the bioavailability of the polysaccharides (Table 1) has remained indescribable due to the inefficient animal metabolism regarding this nutrient. This chemical limitation generally prevents the efficient use of *Ulva* as a single feed for animals [3].

**Table 1.** Nutrient (% dry weight) and mineral (mg 100 g$^{-1}$ dry weight) composition of some edible seaweeds [31].

| Species | Nutrient Composition (%) | | | | | Mineral Composition (mg.100 g$^{-1}$) | | | | |
|---|---|---|---|---|---|---|---|---|---|---|
| | Protein | Ash | Dietary Fiber | Carbohydrate | Lipid | Na | K | P | Ca | Mg |
| Green seaweed | | | | | | | | | | |
| *Caulerpa lentillifera* | 10–13 | 24–37 | 33 | 38–59 | 0.86–1.11 | 8917 | 700–1142 | 1030 | 780–1874 | 630–1650 |
| *C. racemosa* | 17.8–18.4 | 7–19 | 64.9 | 33–41 | 9.8 | 2574 | 318 | 29.71 | 1852 | 384–1610 |
| *Codium fragile* | 8–11 | 21–39 | 5.1 | 39–67 | 0.5–1.5 | - | - | - | - | - |
| *Ulva compressa* | 21–32 | 17–19 | 29–45 | 48.2 | 0.3–4.2 | - | - | - | - | - |
| *U. lactuca* | 10–25 | 12.9 | 29–55 | 36–43 | 0.6–1.6 | - | - | 140 | 840 | - |
| *U. pertusa* | 20–26 | - | - | 47.0 | - | - | - | - | - | - |
| *U. rigida* | 18–19 | 28.6 | 38–41 | 43–56 | 0.9–2.0 | 1595 | 1561 | 210 | 524 | 2094 |
| *U. reticulta* | 17–20 | - | 65.7 | 50–58 | 1.7–2.3 | - | - | - | - | - |
| Red seaweed | | | | | | | | | | |
| *Chondrus crispus* | 11–21 | 21 | 10–34 | 55–68 | 1.0–3.0 | 1200–4270 | 1350–3184 | 135 | 420–1120 | 600–732 |
| *Crassiphycus changii* | 6.9 | 22.7 | 24.7 | - | 3.3 | 5465 | 3417 | - | 402 | 565 |
| *Agarophyton chilense* | 13.7 | 18.9 | - | 66.1 | 1.3 | | | | | |
| *Palmaria palmata* | 8–35 | 12–37 | 29–46 | 46–56 | 0.7–3 | 1600–2500 | 7000–9000 | 235 | 560–1200 | 170–610 |
| *Neopyropia teneraNeopyropia tenera* | 28–47 | 8–21 | 12–35 | 44.3 | 0.7–1.3 | 3627 | 3500 | - | 390 | 565 |
| *Porphyra umbilicalis* | 29–39 | 12 | 29–35 | 43 | 0.3 | 940 | 2030 | 235 | 330 | 370 |
| *Neopyropia yezoensis* | 31–44 | 7.8 | 30–59 | 44.4 | 2.1 | 570 | 2400 | - | 440 | 650 |
| Brown seaweed | | | | | | | | | | |
| *Alaria esculenta* | 9–20 | - | 42.86 | 46–51 | 1–2 | - | - | - | - | - |
| *Eisenia bicyclis* | 7.5 | 9.72 | 10–75 | 60.6 | 0.1 | - | - | - | - | - |
| *Fucus spiralis* | 10.77 | - | 63.88 | - | - | - | - | - | - | - |
| *F. vesiculosus* | 3–14 | 14–30 | 45–59 | 46.8 | 1.9 | 2450–5469 | 2500–4322 | 315 | 725–938 | 670–994 |
| *Himanthalia elongata* | 5–15 | 27–36 | 33–37 | 44–61 | 0.5–1.1 | 4100 | 8250 | 240 | 720 | 435 |
| *Laminaria digitata* | 8–15 | 38 | 37–37 | 48 | 1.0 | 3818 | 11.5–79 | - | 1005 | 659 |
| *L. ochroleuca* | 7.49 | 29.47 | - | - | 0.92 | - | - | - | - | - |
| *Saccharina japonica* | 7–8 | 27–33 | 10–41 | 51.9 | 1.0–1.9 | 2532–3260 | 4350–5951 | 150–300 | 225–910 | 550–757 |
| *S. latissima* | 6–6.26 | 34.78 | 30 | 52–61 | 0.5–1.1 | 2620 | 4330 | 165 | 810 | 715 |
| *Sargassum fusiforme* | 11.6 | 19.77 | 17–69 | 30.6 | 1.4 | - | - | - | 1860 | 687 |
| *Undaria pinnatifida* | 12–23 | 26–40 | 16–51 | 45–51 | 1.05–45 | 1600–7000 | 5500–6810 | 235–450 | 680–1380 | 405–680 |

### 2.2. Red Seaweeds

In general, compared to green and brown algae, red algae contains a high amount of proteins (Table 1) reaching 47% (*Neopyropia tenera*) of a dry matter [32].

The proteins from this seaweed group are made up of one or more chains of amino acids, especially glycine (Gly), alanine (Ala), arginine (Arg), proline (Pro), glutamic (Glu) and aspartic (Asp) acid (compose large part of the amino acid fraction), whereas tyrosine, methionine and cysteine appear in a lower quantity. Glutamic and aspartic acid, that have acidic side chains at neutral pH, in red seaweeds represent 14–19% of amino acids [33]. Dawczinski et al. [11] found relevant values of the amino acid taurine (tau) in red seaweeds unlike the brown seaweeds. Essential amino acids reveal almost half of the total amino acids and their protein profile is close to the egg's protein profile. In general, all algae have the same amount of nonessential amino acids [11].

Lipids, in these seaweeds present relatively lower contents, 1–5% of dry matter (Table 1), found in *Chondrus crispus* (1.0–1.3%) and *Palmaria palmata* (0.7–3%) [27,34]. Red seaweed predominantly contains the polyunsaturated 20 carbon-fatty acids eicosapentaenoic acid (EPA, ω-3, C20:5) and arachidonic acid (AA, ω-6, C20:4). Palmitic acid (C16:0) is the main saturated fatty acid with 26% and monounsaturated is oleic acid [11]. *Neopyropia/Porphyra/Pyropia* sp. were tested and the assays showed that palmitic, eicosapentaenoic, arachidonic, oleic, linoleic and linolenic acid were the main fatty acids [11]. Another class of lipids are sterol compounds. Most red algae contain cholesterol, desmosterol, sitosterol, fucosterol and chalinasterol [35].

In their study, Dawczinski et al. [11] did not find significant differences between the red and brown algae, as they both revealed low fat and high fiber content (Table 1). Red algae contains soluble fibers such as sulphated galactans (agars and carrageenans), xylans and floridean starch [11,35,36].

Red algae (Rhodophyta) are seaweeds with an interesting nutritional profile. The minerals present in some red algae, namely *Chondrus crispus* and *Gracilariopsis* sp., are Na, K, Ca and Mg, as well as, Fe, Zn, Mn and Cu [36]. The iodine content in red algae is high, in *Gracilaria* sp. reaching 426 mg/100 g of seaweed dry biomass but not as high as in brown algae (1200 mg/100 g of seaweed dry biomass). Algae iodine has already contributed to the nutritional enrichment of the meat of several fish species [37].

Red seaweed outperformed several brown and green seaweeds in sequestering negatively charged hexavalent chromium ions. It possesses more cationic sites, which show low affinity for positively charged metal ions, such as cadmium, but higher affinity for hexavalent chromium [10].

Algae may have bioactive compounds and bioactive secondary metabolites. Red algae are the main source of halogenated monoterpenes. Seaweed contains also sesquiterpenes, diterpenes, C15-Acetogenins, C27 and C28 steroids, with C29 steroids, in small amounts [38].

Most red seaweeds contain water-soluble vitamins B and C (Table 2), mainly amine and riboflavin and liposoluble vitamins such as carotenoids (as provitamins of vitamin A). The carotenoids are represented by different pigments which form the resulting seaweed color together with chlorophyll and are also very strong antioxidants. The main carotenoids of red seaweed are α-and β-carotene and their derivates such as zeaxanthin and lutein [10,36].

**Table 2.** Vitamins contents (mg 100 g$^{-1}$ edible portion) of seaweeds [31].

| Species | \multicolumn{10}{c}{Vitamins (mg 100 g$^{-1}$)} |
| --- | --- | --- | --- | --- | --- | --- | --- | --- | --- | --- |
| | A | B$_1$ | B$_2$ | B$_3$ | B$_5$ | B$_6$ | B$_8$ | C | E | Fatty Acids |
| \multicolumn{11}{c}{Green seaweed} |
| *Caulerpa lentillifera* | - | 0.05 | 0.02 | 1.09 | - | - | - | 1.00 | 2.22 | - |
| *C. racemosa* | - | - | - | - | - | - | - | - | - | - |
| *Codium fragile* | 0.527 | 0.223 | 0.559 | - | - | - | - | <0.223 | - | - |
| *Ulva compressa* | - | - | - | - | - | - | - | - | - | - |
| *U. lactuca* | 0.017 | <0.024 | 0.533 | 98 * | - | 6 * | - | <0.242 | - | - |
| *U. pertusa* | - | - | - | - | - | - | - | 30–241 ** | - | - |
| *U. rigida* | 9581 | 0.47 | 0.199 | <0.5 | 1.70 | <0.1 | 0.012 | 9.42 | 19.70 | 0.108 |
| *U. reticulta* | - | - | - | - | - | - | - | - | - | - |
| \multicolumn{11}{c}{Red seaweed} |
| *Chondrus crispus* | - | - | - | - | - | - | - | 10–13 * | - | - |
| *Crassiphycus changii* | - | - | - | - | - | - | - | 16–149 ** | - | - |
| *Agarophyton chilense* | | | | | | | | | | |
| *Palmaria palmata* | 1.59 | 0.073–1.56 | 0.51–1.91 | 1.89 | - | 8.99 | - | 6.34.34.5 | 2.2–13.9 | 0.267 |
| *Neopyropia tenera* | - | - | - | - | - | - | - | - | - | - |
| *Porphyra umbilicalis* | 3.65 | 0.144 | 0.36 | - | - | - | - | 4.214 | - | 0.363 |
| *Neopyropia yezoensis* | 16,000 *** | 0.129 | 0.382 | 11.0 | - | - | - | - | - | - |
| \multicolumn{11}{c}{Brown seaweed} |
| *Alaria esculenta* | - | - | 0.3–1 * | 5 * | - | 0.1 * | - | 100–500 * | - | - |
| *Eisenia bicyclis* | - | - | - | - | - | - | - | - | - | - |
| *Fucus spiralis* | - | - | - | - | - | - | - | - | - | - |
| *F. vesiculosus* | 0.30–7 | 0.02 | 0.035 | - | - | - | - | 14.124 | - | - |
| *Himanthalia elongata* | 0.079 | 0.020 | 0.020 | - | - | - | - | 28.56 | - | 0.176–0.258 |
| *Laminaria digitata* | - | 1.250 | 0.138 | 61.2 | - | 6.41 | 6.41 | 35.5 | 3.43 | - |
| *L. ochroleuca* | 0.042 | 0.058 | 0.212 | - | - | - | - | 0.353 | - | 0.479 |
| *Saccharina japonica* | 0.48 | 0.2 | 0.85 | 1.58 | - | 0.09 | - | - | - | - |
| *S. latissimi* | 0.04 | 0.05 | 0.21 | - | - | - | - | 0.35 | 1.6 | - |
| *Sargassum fusiforme* | - | - | - | - | - | - | - | - | - | - |
| *Undaria pinnatifida* | 0.04–0.22 | 0.17–0.30 | 0.23–1.4 | 2.56 | - | 0.18 | - | 5.29 | 1.4–2.5 | 0.479 |

* expressed as ppm; ** expressed as mg%; *** expressed as I.

### 2.3. Brown Seaweeds

In general, brown algae (Phaeophyceae) are seaweeds with the lowest protein content (Table 1), when compared to red and green algae. The most frequently determined protein content in brown seaweeds occurs within a declared range of 5 to 15% [9,10].

The concentrations of EAA in brown algae differ substantially between species. The concentrations of threonine (Thr), valine (Val), isoleucine (Ile), leucine (Leu), phenylalanine (Phe), lysine (Lys) and methionine (Met) were higher in *Undaria pinnatifida* than in *Laminaria* sp.; however, *Laminaria* sp. had higher concentrations of cysteine (Cys) than *Undaria pinnatifida*. Aspartic acid and glutamic acid amino acids were the most abundant in these algae species tested in this study [10]. Brown algae contained higher concentrations of phosphoserine than red algae [10] while glutamic and aspartic acid represents 20–44% [32].

The type of carbohydrate varies greatly among the algae. The soluble fibers are alginates, fucans and laminarins for brown seaweeds. Fucoidans, sulphated polysaccharides, are extensively involved in the cell walls of brown seaweed [29]. In terms of dietary fiber (Table 1), it is not uniform in all brown algae [10]. Fucoidans present several physiological and biological characteristics, such as antitumor, anticoagulants, antioxidant, antiviral and antithrombotic activities, besides the impact on the inflammatory and immunological systems [28].

According to some researchers, laminarin is the second main source of glucan in brown algae and it was detected as a regulator of intestinal metabolism through its impact on mucus structure, intestinal pH and short chain fatty acid formation [29].

Brown algae are balanced sources of omega-3 and omega-6 acids [29,36]. The brown seaweeds, such as *Undaria pinnatifida*, *Laminaria* sp. and *Hizikia fusiforme*, contain predominantly 20 polyunsaturated eicosapentaenoic acids (EPA, ω-3, C 20:5) and arachidonic acid (AA, ω-6, C 20:4) [10]. Palmitic acid (C16:0) is one of the most abundant but not as abundant as in most red algae. In general, other fatty acids, abundant in brown algae, are the essential fatty acids, oleic acid (ω-9, C18:1), linoleic acid (ω-6,C18:2), linolenic acid (ω-3,C18:3) and the precursors of the eicosanoids, arachidonic acid (ω-6,C20:4) and eicosapentaenoic acid (ω-3,C20:5) [36]. Some saturated and monounsaturated fatty acids are found in abundance (Table 2) only in some brown algae (*Laminaria* sp. and *Undaria pinnatifida*) such as myristic and palmitolenic acid [10].

Polyphenols (fucol, fucophlorethol, fucodiphloroetol G ergosterol) and the phenolic compound phlorotannin are also abundant, in *Sargassum*, *Fucus* and *Ascophyllum nodosum*, and they have strong antioxidant effects. These seaweeds also contain halogenated compounds [31].

The minerals present in some brown algae (Phaeophyceae), namely *Undaria pinnatifida* (Table 1) and *Sargassum* sp., are some of the main ones (Na, K, Ca and Mg), as well as Fe, Zn, Mn and Cu [29]. In certain brown algae the concentration of iodine can reach very high levels, in particular the genus *Laminaria*. According to several authors, *Saccharina japonica* (as *Laminaria japonica*) presented the highest iodine content of 5.6 and 3.04 mg/kg among other seaweeds studied [9,31].

Brown algae can participate in the accumulation of metals due to their carboxyl groups and because the cell wall is formed by cellulose. They have cationic characteristics but less than red algae [9].

Brown algae contain considerably higher concentrations of arsenic than red or green algae. In most species of seaweeds, the concentrations are below 54 mg kg$^{-1}$ dry weight, and 5–10% of the total arsenic is organic [38].

Some of the most important vitamins present in most brown algae are vitamin C, E and group B vitamins (Table 2), especially thiamine and riboflavin [9]. Vitamin B12 is present in brown algae in lower concentrations than red and green algae [29]. Brown seaweed contains larger amounts of vitamin E and high amounts of vitamin C. Brown seaweed carotenoids are formed by fucoxanthin, β carotene, lutein, violaxanthin, antheraxanthin, zeaxanthin and neoxanthin. Fucoxanthin is the main carotenoid in brown seaweed and has been shown to have many health benefits [9].

## 3. Seaweed as Valuable Nutritional and Nutraceutical Animal Feed

The nutritional value attributed to macroalgae along with their nonanimal nature makes them particularly appropriate to be used in animal feed as nutraceuticals, a term that results from the combination of nutritional and pharmaceutical, used to identify food components that bring health benefits, including the prevention to some diseases [39,40].

The health benefits of seaweed, beyond the provision of essential nutrients, have been supported by in vitro studies and some animal studies; however, many of these studies have inappropriate biomarkers to substantiate a claim and have not progressed to suitably designed trials to evaluate efficacy. The limited evidence that does exist makes some seaweed components attractive as functional food ingredients, but more animal nutritional studies evidence (including mechanistic evidence) is needed to evaluate both the nutritional benefit conferred and the efficacy of purported bioactivities and to determine any potential adverse effects [41]. Through an evaluation of the nutritional composition of edible seaweeds in Section 2, this section summarizes the available evidence and outlines the potential of the seaweeds as animal feed hypothesis with a prominent feed safety question.

### 3.1. Feed Safety

The animal feed plays a vital role in the global food security, and it is conceived to ensure the sustainable production of safe and affordable animal proteins. With the increase of the animal production, it will be necessary for more feed to be produced, which will be safety certified. Consequently, new and old feed sources are being controlled for hazards and critically analyzed to guarantee feed safety for animal consumption. However, the food safety regulatory framework is not fully harmonized between the countries, creating a lack in feed safety chain, increasing the animal health risks and the animal consumption by the humans [42].

Seaweed are considered a rich and sustainable source of macronutrients (particularly dietary fiber) and micronutrients to the animal feed, but if seaweeds are to contribute to future global food security, legislative measures to ensure monitoring and labeling of feed products are needed to safeguard against excessive intakes of salt, iodine, and heavy metals, such as arsenic (As), aluminum (Al), cadmium (Cd), lead (Pb), rubidium (Rb), silicon (Si), strontium (Sr) and tin (Sn) [43,44]. While heavy metal concentrations in seaweeds are generally below toxic levels, bioaccumulation of arsenic and lead are the main risk in wild seaweed harvest, and more studies of heavy metal toxicokinetics are needed to address the problem.

Levels of arsenic, mercury, lead, and cadmium in 426 Korean dried seaweed products ranged from 0.2 to 6.7% of provisional tolerable weekly intakes when 8.5 g of seaweed was consumed per day in human food consumption [41]. Chen et al. [43] revealed the different levels of Al, As, Cd, Cr, Cu, Hg, Mn, Ni, Pb and Se in dried seaweeds from southeastern China (Zhejiang province). This indicates that element concentration changes with different species of seaweeds and origin areas. For example, the levels of Cd, Cu, Mn and Ni in red seaweeds (*Porphyra*) were significantly higher than those in brown seaweeds (*Laminaria*, *Saccharina* and *Undaria*).

A tradeoff between iodine and/or heavy metal ingestion and the amount of whole seaweed needed to obtain meaningful amounts of PUFAs, protein or dietary fiber may limit the recommended portion size of the seaweeds concentration in feed [41]. Relevant and key information to use seaweeds with feed safety guarantee will be gathered. However, for most countries there is no regulation on maximum levels of heavy metals in seaweed [43]. However, there is Regulation (EC) No 1831/2003 laying down rules governing the European Community authorization of feed additives. In addition, Regulation (EC) No 429/2008 lays down detailed rules for the implementation of Regulation (EC) No 1831/2003 as regards the preparation and the submission of applications and the assessment and the authorization of feed additives [45].

The production of rancid flavors and odors due to oxidative stress can lead to a reduction in the sensory attributes, nutritional quality and food safety. Extracts from seaweeds are rich in polyphenolic compounds which have well documented antioxidant properties. They also have antimicrobial

activities against major food spoilage and feed pathogenic micro-organisms. The addition of seaweeds or their extracts to feed products will reduce the utilization of chemical preservatives, which will fulfill the industry as well as consumer demands for "green" products. In addition, the current status and the future projections in the functional effects of seaweeds as a means to improve the fiber content and reduce the salt content of food products will be of significant importance to the meat industry [17].

### 3.2. Fish Farming

Over 50% of the operating costs in an intensive fish aquaculture are related to the fish feeding [46,47]. This led to research, in recent decades, for new sources of aquaculture nutrients, especially on terrestrial plants as legumes and oilseed crops [47–49]. Algae are a natural alternative to soybean for fish diets, presenting economic and nutritional advantages, since the nutritional profiles made to soybean show that this plant does not fully match the fish nutritional requirements [47,50,51]. This problem has been attributed to using a certain amount of plant protein sources which might contain antinutritional factors and result in palatability problems [47,52,53].

In recent years, there were some experiments carried out with the objective to find more economically and nutritional viable options for fish feed. Al-Hafedh et al. [54] experimented with the application of green *Ulva* and red seaweeds *Gracilaria* with the objective to reduce nutrient concentrations in sea water effluent, sources of pollution and to diversify origin of the feeds in the changing market status as offering the possibility for additional sources of income. This was considered to be highly relevant to develop the industry of aquaculture [55] and to reduce the dangers of an oligotrophic sea that has a high level of biodiversity [47,56,57]. The Integrated Multitrophic Aquaculture (IMTA) is a cultivation system that is based in the fish or shrimp (herbivores or carnivores) aquaculture with cultivation of others species, such as sea urchins or bivalves and seaweeds to capture the excess of nutrients from the effluents of the fish or shrimp tanks or cages. Basically, the seaweeds (inorganic nutrients absorber) "clean" the nutrient-enriched water output from fish or shrimp aquaculture, thus providing a possible source for the aquatic animal feed. The IMTA provides sustainable conditions to an intensive culture of fish or shrimp, which is being practiced in many countries, as integrated unit with seaweeds and mollusk culture. This approach, besides being a form of balanced ecosystem management, prevents potential environmental impacts from aquaculture. It also provides exciting new opportunities for valuable crops of seaweeds, transforming it in one of the fields for further technological advances in seaweed aquaculture [58].

Other experiences with diverse seaweeds, such as *Ulva* sp., *Neopyropia/Porphyra/Pyropia* sp., *Gracilaria* sp., *Ascophyllum nodosum*, *Sargassum* sp. and *Padina* sp. led to encouraging results for the use of seaweed as fish feed, which appears to depend on seaweed species, its incorporation level and the fish species where the seaweed is produced [59–68]. In short, it was discovered that using seaweed meals as supplement to fish diets enhance the growth, lipid metabolism, physiological activity, stress response, disease resistance and carcass quality of various fish species [69–71].

In 2019, Kamunde et al. [72] studied a salmon meal based on brown seaweed *Laminaria* sp. named AquaArom®. The protocol was intended to access food intake, growth performance, plasma antioxidant capacity and mitochondrial respiration. The conclusions showed that the addition of AquaArom® to commercial salmonid food improves those characteristics and alleviates the effect of temperature rise on mitochondrial respiration. The slight decline in crude protein and minerals resulting from the addition of up to 10% AquaArom® to aquafeed appear to have no adverse consequences on Atlantic salmon smolts. Thus, mixing of brown seaweed meal with commercial aquafeeds (and potentially feeds from other farm animals) could offer a cost-effective way of harnessing the beneficial effects of seaweeds in animal production. The lower level of protein accessed in this experiment, resulted from the use of brown seaweed (known for low crude protein levels), and it may be complemented by green or red seaweed [72]. Other brown gigantic seaweeds (commonly, named kelp), *Saccharina latissima* (Phaeophyceae) demonstrated potential to be inserted as feed additive (with concentration below 4%) for rainbow trout, where the fatty acids (e.g., oleic and linoleic acids) and lipid fraction in the fillet

was reduced, although the eicosapentaenoic acid and docosahexaenoic acid (omega-3 PUFAs) was not affected. Kelp supplementation also increased a protective activity against oxidative stress in this fish [73], being in line with the data gathered about kelp, which points out that, as a brown seaweed, it produces secondary metabolites with more acceptable nutraceutical characteristics.

In conclusion, there is a great quantity of data to support the use of seaweed in order to achieve a higher productivity on fish farms. The advantages range from growth and development rates, disease resistance, financial gain and even ecological preservation. This is a case where the implementation of such research and development (R&D) was a relatively easy process, leading to various companies developing their own products, as exemplified earlier with AquaArom®, proving, once again, the safety of using seaweed as fish feed.

### 3.3. Oyster Feed

Oyster is a highly valorized and appreciated seafood product. It is one of the most widely cultivated marine animals that in 2014 exceeded 600,000 tones worldwide [55].

Production of oysters in hatchery encompasses three distinct stages: broodstock conditioning, larval culturing and postlarval rearing phase. Broodstock conditioning is a stage of the utmost importance in hatchery because it is the first chance to modulate/condition the whole offspring. A higher oyster fecundity, better quality eggs and enhanced larval viability are possible through intelligent innovations in broodstock conditioning [74,75]. It has been shown that physical and nutritional factors can modulate gonadal development, either accelerating gametogenesis or slowing gonadal maturation [76]. Several nutritional studies were made on bivalve species that have tested different algal compositions for an enhanced reproductive outcome [74,77–80]. Currently, the most successful strategy to a balanced diet lies in microalga blends; due to its nutritional value, it is possible to obtain optimal food conditions [81]. This entails a dependence on the production of live microalgae, which may represent up to 30–40% of the hatchery operational costs [82] but can show an economic limitation for the use of such a diet in bivalve hatcheries [83,84]. This leads to a new research line focused on seaweed—it can be a nutritional source for oyster, human and animals [27].

Nutritional profile influences the physiology of bivalves, having a strong effect on their proteins, carbohydrates and lipids [85,86], being that these organisms are known to be mineral accumulators. For example, bivalves are rich in potassium, sodium, calcium, magnesium, phosphorus and are a source of essential trace elements, such as iron, manganese, copper, zinc, cobalt, selenium and iodine [87]. Some species of green seaweed, such as *Ulva* sp., have high protein contents (10–25% in dry content) and high levels of mineral elements with nutritional value, including calcium and magnesium [88]. This suggests the hypothesis of seaweed use as feed, at least partially, on oyster and other bivalve feeds; moreover, they have been shown to improve stress response and resistance to disease, thereby representing a meaningful advantage to aquaculture [88].

Before further advancing, there is a fundamental question that has to be addressed since seaweeds (for example *Ulva* sp. *and Fucus* sp.) are also known to be mineral element accumulators, some of them highly toxic for humans, for example iodine, arsenic and mercury [89]. As seaweed can accumulate highly toxic minerals, so can oysters, using seaweeds as food, that can represent a way to introduce hazardous elements into the oyster feeding and, respectively, human food. Regarding this, in order to evaluate the potential nutritional value of oysters, it is crucial to know whether any given nutrient, as part of feed, can or cannot be bio-accessible for humans [89,90].

Cardoso et al. [90] designed an experimental protocol to determine better macro and microalgae blend to feed pacific oyster (*Crassostrea gigas*) that describes the bio-accessibility of nutrients and minerals. It was observed that oysters consuming only one seaweed species, independent from species studied, had the highest levels of Be, Cu, Zn, Sr and Cd. The most important problem in the oysters' composition is the increment of microalgae concentrations in the oysters' feeding system with a progressive concentration reduction of seaweed. When high levels of Cd or Pb were found or Zn in oysters, the study indicates that caution and further study are needed to guarantee and maintain

low the heavy metal levels during the substitution of the feed source. It was also observed that Mn, Cd and Pb bio-accessibility has increased with the substitution of the initial microalgal with seaweed feed, proving the seaweed potential as the oyster feed with reduction of the above cited dangerous elements [90].

As demonstrated in this topic, there is promising data to support the use of green seaweed, mainly *Ulva* sp. in order to achieve a higher productivity of oyster farms due to high protein contents and high levels of mineral elements found in oysters. These advantages can also result from improved stress response, resistance to disease, financial gain and ecological preservation. However, a capacity of metal accumulation, some of them toxic, shared between seaweed and oysters, raise some questions, which must be taken into account during the development of the oyster feed product.

### 3.4. Poultry Feeds

When addressing the issue of poultry farming, it is necessary to distinguish between poultry raised for the consumption of their meat (broiler poultry) and poultry raised for egg consumption (laying poultry). Thus, subjects will be addressed in two different points.

There are three main reasons to use seaweed in poultry feed: to improve animal immune status, to decrease microbial load in the digestive tract and for beneficial effects on the meat and eggs [2,89,91–95].

### 3.4.1. Broiler Poultry

In broiler aviaries, feed is based primarily on corn and soybean meals, with corn in most parts of the world as a source of energy due to its abundance and digestibility (60–75% of broilers diet). Historically, high corn prices led to the search for new feed capable to provide the required nutrients for broilers, in order to maintain productivity and lower the feed price.

Green algae (Chlorophyta) have been studied in the previous century as an alternative to feed poultry. Asar [96] found that supplementation of chicken's basal diet with 4% of seaweed increased body weight gain. El-Deek et al. [97] found no significant effects on growth, feed intake and feed conversion ratio with the inclusion of *Sargassum* ssp. (Phaeophyceae) on broiler diet. Gu et al. [98] concluded that a 2% seaweed inclusion on the broiler feed improved performance and dressing percentage. Ventura et al. [99] compared animal feeds with two different concentrations of *Ulva rigida* (10 and 20%); the data obtained indicated that the feed intake and body weight gain was better with 10% of *U. rigida*. The feed with 20% of *U. rigida* had a harmful effect on the broilers. Later, it was found that poultry fed with 10% mixture of green algae, containing various species as genus *Ulva*, *Caulerpa*, *Codium*, *Halimeda* and *Bryopsis*, showed better growth with statistical differences in body weight, a lower level of fats (0.7–1.7%) and higher protein contents (46.6–72.2%) when compared to control groups (1.1–3.2% and 66.4–71.4%, respectively). Species like *Codium* sp., with spongy thallus, can retain high amounts of salt in this structure which can lead broilers to lose weight because of diarrhea [100].

Studies with another *Ulva* species, *U. lactuca* (Sea lettuce), which can be found on Atlantic shores, pointed out that, with lower than 3% seaweed added to animal diet, broilers performed better than the respective control diet. It was speculated that higher crude protein and amino acids, especially methionine, plays an important role in the improvement of dressing and breast yield. However, as mentioned before, seasonal variations in nutrient composition of seaweed must be considered [93]. With this result, *U. lactuca* can be pointed out as a more economical ingredient to be incorporated, at least partially, in broiler feeding.

The nutraceutical characteristics of seaweed have been the subject of studies in recent years. The work of Kulshreshtha et al. [101] included red seaweeds (*Chondrus crispus* and *Sarcodiotheca gaudichaudii*) in livestock feed with the objective to study its nutritional value and the prebiotic potential. The research started with the fact that antibiotics are used to stimulate growth and to control disease-causing pathogens in layer chickens [101–104]. The prolonged and indiscriminate use of antibiotics in livestock led to concerns such as development of antibiotic-resistant strains of

pathogens, high concentrations of antibiotic residues in meat and meat products and undesirable changes in the microbial communities of animal gastrointestinal tracts [101,105–107] and as a result, numerous countries, including the UE, banned the use of antibiotics as a growth promoter. There was not significant improvement found by joining red seaweed in broiler feed in parameters as growth, development, feed intake or egg production, when compared to control [101]. However, the inclusion of red seaweed is more interesting from the prebiotic point of view. In fact, this research found that the seaweed species included in the meal showed an increase in the population of beneficial bacteria and a reduction of pathogenic bacteria in the gut, improvement in villi height, crypt depth, and an increase in the concentration of short chain fatty acids, which can also be replicated in laying hens [101,108]. These results are directly linked to the fact that red algae contain specific bioactive compounds, such as agars, carrageenans, xylans, sulphated galactans and porphyrins that may be responsible for the effects [109–111]. However, further study is needed to determine such mechanisms.

As a practical conclusion, and based in what was described, it appears to be possible to enrich broiler feed with green seaweed, or a mixture of green and red seaweed, in order to stimulate both the growth and the health of the broilers. The limiting factor appears to be the use of low concentration (1–2%) of seaweed in the meal, which could represent a healthier method to achieve the proposed objectives, as well as, a less expensive one, when compared to the actual methods [111]. Following this data, the next logical step should be the investment in R&D work in order to create products, based on seaweeds, able to be included in the market as an alternative to the existing ones.

### 3.4.2. Laying Poultry

Eggs are one of nature's most wholesome foods because of their content in essential and nonessential minerals, high-quality proteins, lipids and vitamins. Egg composition can be altered by hereditary genes, diet and poultry age. Egg yolk contains natural carotenoids, and its yellow color is attributed to β-carotene, zeaxanthin, kryptoxanthin and lutein, which are easily found in commercial feed [39,112–114]. Alongside those carotenoids, eggs represent an important source of protein, minerals (phosphorus, iron) and easily digestible fats (93–96%) like ω-3 fatty acids, all of which can be enriched by supplementation of the poultry feed [40].

There was research in the last decade with the objective to enrich the egg molecular content and to adjust it for better human consumption. This can be achieved by supplementation of the poultry feed with seaweed, which can be used to enhance the levels of vitamins, minerals and fatty acids, mainly ω-3 fatty acids [115–117].

Research was done with green algae from the genus *Ulva*, with the inclusion of 1–3% of this seaweed resulting in improved egg production and quality, increasing the weight, shell thickness, yolk color and reduced yolk cholesterol. The seaweed extract also reduced *Escherichia coli* load in feces, which suggests better health of the animals and a decreasing feed conversion ratio [2,95]. These results need further studies in order to access the bioavailability of seaweed contents in order to determine what concentration is the best.

Recently, red seaweed, such as *Chondrus crispus*, has been used at 2–4% feed to reduce the level of *Salmonella enteritidis*, a toxic bacterium which can be transmitted vertically from laying hens to eggs through the ovaries and oviducts or due to contaminated feces. Reducing the level of this bacterium is vital to produce safer eggs and reduce the spread of salmonellosis to humans. A greater reduction of negative effects was observed on layer growth and egg production caused by *S. enteritidis* with this seaweed at the 4% diet. Dietary inclusion of *C. crispus* inhibited colonization of Salmonella in the excreta and ceca, which could be due to promoting the growth of *Lactobacillus* and increasing the concentration of short chain fatty acids. A higher level of IgA in birds supplemented with this feed indicates a direct role of seaweed on the maturation of the humoral immune system. These results encourage the research for a nontoxic alternative feed that producers (including organic farmers) can accept [108].

Using brown seaweed has the potential, like *Sargassum* sp., at a 3–6% dietary level, to give benefits to the egg quality, decreasing yolk cholesterol, triglycerides and ω-6 fatty acids and increased carotene and lutein plus zeaxanthin contents. There are data of poultry being fed with boiled seaweed which resulted in improvement of the high density lipoprotein, which is beneficial for human health [118]. The research even approached the way in which seaweed should be used as feed to layers, designing a protocol in which groups should be applied as sundried, boiled or autoclaved seaweed at 3 and 6%. This approach intends to expand further on which edible brown seaweed offers a variety of health benefits, mainly due to the relatively high contents of ω–3, Ca and Fe. Results have shown that there are differences between the groups. However, it was concluded that those differences could be due to geographical location, year season, environmental factors, growth media and physiological conditions [118]. Further research should include controlled production of seaweeds (in aquaculture, for example) in order to maintain a seaweed stability profile and minimize the influence of such external factors. In this way it should be possible to determine how laying poultry feed should be administered in order to maximize its advantages.

As can be concluded, the use of various seaweed species (being green, red or brown) has the potential to enhance various qualities on poultry eggs. Such as quality, weight, yolk cholesterol reduction and, depending on the species, other bioactive molecules capable even of reducing toxic bacterium levels in the digestive system of poultries. It appears that a mixture between brown, green and red seaweed could be a promising supplement used in order to enrich eggs. However, such a product would need R&D work in order to determine the bioavailability of the molecules in a seaweed mixture. Once again, the concentration of the seaweed in the feed seems to be crucial.

### 3.5. Ruminat Feed

The use of seaweed in ruminant feeds has been affected by the high demand of animal feed protein, the need for alternatives to the traditional soybean and animal protein feed as well as the food market regulations related with the livestock feeding. Studies carried out to date regarding the use of seaweed in bovine, caprine and other ruminant nutrition have focused on the addition of small quantities of different macroalgal species to the feed and the subsequent assessment of the animal to check for possible prebiotic activity and enhanced animal performance.

Information on the application of green seaweeds in ruminant feed is scarce. *Ulva lactuca* could be fed to male lambs at up to 20% of diet, without negatively affecting the palatability. It presents low protein degradability (40%) and a moderate energy digestibility (60%), being comparable to a medium to low quality forage and suitable to use with feeds that have high energy/low protein content as cereal grains [119]. *Chaetomorpha linum* (Chlorophyta) was also used to feed growing lambs, with a 20% seaweed meal, having a slightly depressing effect on growth and feed conversion ratio, possibly due to the high ash content [9,120].

Red seaweed has received more attention, as demonstrated before, in bovine feed than in other ruminant feed [9]. There are some uses of red seaweed (a 70% concentrate of *Phymatolithon calcareum*—as *Lithothamnion calcareum*—extract fed at a ratio of 0.5 g/kg) with success in buffering the rumen pH, but they did not improve fiber digestion nor modify rumen fermentation [9,121]. This is in agreement with the literature, since the genus *Ulva* presents low ash levels (Table 1), which allows this seaweed to become a great option for future studies with bioavailability.

For example, supplementation of the brown seaweed *Ascophyllum nodosum* to feedlot cattle was found to reduce fecal shedding of *Esherichia coli* [2,121]. There is more research with the inclusion of seaweed in caprine feed. Orkney sheep, from the North Ronaldsay Island, are known to feed mostly brown seaweed most of the year. Species like *Laminaria digitata*, *Laminaria hyperborea* and *Saccharina latissima* (Phaeophyceaea) accounts for 90% of the summer feed of this sheep, meeting a substantial amount of nutrient requirements since they may have up to 13% crude protein. Orkney sheep also consume another seaweed species, like *Alaria esculenta*, *Ascophyllum nodosum*, Fucus sp. (brown seaweed), *Palmaria palmata* (red seaweed) and some green algae. Sheep consume

seaweed in such quantity to sustain maintenance requirements but suffer from mineral overload due to its high mineral content [2,122]. There are also some studies suggesting the use of *Macrocystis pyrifera* up to 30% levels as a supplement in goat feed without affecting digestibility, degradability and parameters of ruminal fermentation (such as pH and ammoniac nitrogen). It was also noticeable the increase of rumen pH, water intake and urine excretion [2,122]. Species from the Genus Sargassum are also studied for this purpose. Nowadays, we know that it could be introduced at up to 30% in the diets of growing sheep and goats without depressing intake, growth performance and diet digestibility [59,123,124]. Eating *Sargassum* sp. increased water consumption, probably due to their high concentration in minerals, mainly Na and K, which could make *Sargassum* less suitable for feeding during dry periods. *Sargassum* sp. meal could be used to limit the decrease in rumen pH resulting from acidogenic diets. It also tended to decrease the concentration of volatile fatty acids [124]. Further research can incorporate the determination of bioavailability in a mixture between *Ulva lactuca* and one or more of the options mentioned in terms of red seaweed. This will allow us to understand if such a mixture can be used as a prebiotic, retaining the advantages of both species present in the mixture.

There are various observations of using mainly brown and red seaweed as ruminant feeds. However, the data is scarce with the exception of few punctual cases and is not enough to start R&D work in order to develop new products for the ruminant feed market. There are a lot of studies to be developed in order to sustain seaweed as feed supplement in ruminants.

*Asparagopsis armata*: The Future for Methane Emissions Reduction from Ruminant Animals?

The red seaweed *Asparagopsis armata* is one of the best hypothesis exploited to ameliorate one of the main problems that livestock farms face nowadays: high rates of enteric methane emissions [125]. Enteric methane is a natural by-product of microbial fermentation of nutrients in the digestive tract of animals [126]. There are considerable differences in contribution of enteric methane in different regions and countries of the world. For instance, it was estimated in 2017 that the enteric methane emissions from livestock, the main source of anthropogenic methane emissions in the US, reached 6.46 million tons, which is equivalent to 27% of the nation's anthropogenic emissions [127,128].

Seaweed has been a traditional part of the livestock diet and they have a historical usage in agriculture [1]. There have been several studies on seaweeds to characterize their effects as livestock feeds and their potential to manipulate rumen fermentation and methane production, which determined that the formulation of the basal feed is of key importance. Many seaweed species have been demonstrated to reduce methane production by rumen methanogens but with variable effects on fermentative health and substrate digestibility [129]. The *A. armata* is the only seaweed that demonstrated to remain effective and dramatically anti-methanogenic without negative impacts on rumen function and, at low inclusion levels, in animal diets [126,130,131]. Most of the initial breakthroughs in the inclusion of *A. armata* as livestock feed occurred in in vitro studies, all of which have demonstrated significant reduction of methane emissions at levels of approximately 2% of diet substrates [132–134]. Although it was considered that this dietary level of the seaweed was low and considered feasible for livestock production systems, in 2018 it was proved to be potentially effective at lower intake levels. Their study in sheep using *Asparagopsis taxiformis* reported up to 80% reduction of methane emission. This research was also important because of the observation of the refusal of the tested animals in the assay to ingest meals with high levels of seaweed, proving the potential of low intake levels [18]. The potential of the seaweed *Asparagopsis armata* to reduce methane emissions shown in in vitro studies was recently investigated in vivo using lactating dairy cows, thus evaluating the methane emission alongside the impact of seaweed on the quality of the milk produced. Adding *A. armata* at 0.5% reduced the methane production by 26.4%, by 20.5% in methane yield (adjusted for feed intake) and by 26.8% in methane intensity (adjusted for milk production) without compromising milk yield or intake. Increasing to a level of 1% resulted in reductions of 67.2% methane production, 42.6% methane yield and 60.0% methane intensity. However, feed intake and

milk yield were also reduced. Bromoform concentration in milk was not significantly different in cows that consumed seaweed compared to control. Other mineral concentrations in milk may be increased so some processing may be necessary for *A. armata* to be used as a feed additive [126].

The work of Kinley et al. [135] demonstrated the effectiveness of *Asparagopsis* sp. in beef cattle fed with a high grain diet. The conclusions of this work point out *Asparagopsis* sp. included in the diet at 0.05, 0.10 and 0.20% and resulted in decrease of methane production (g/kg dry matter intake) of 9, 38 and 98%, respectively. Enteric $H_2$ emissions increased with increasing *Asparagopsis* inclusion by 0, 380 and 1700% without compromising feed intake. Growth rate of the steers was enhanced by the 0.10 and 0.20% inclusion levels after 90 days finishing period with average daily weight growth increases of 26 and 22%, respectively. Including *Asparagopsis* sp. in the concluding 60 days those values were enhanced by 51 and 42%, respectively.

This demonstrates that *Aspargopsis* sp. can be a player key for reducing the methane emission in the ruminants, without secondary effects; however, the seaweed needs to be added to the normal feed as feed supplement to be effective. The effect of *A. armata* is not only methane reduction; it can also supply important minerals for the ruminant growth and dietary digestibility, but the data regarding the last one is scarce, thus there is a need for more studies in this area. One of the main topics which should be studied is related with the bioavailability of metabolites and minerals during the livestock digestion, being that this topic is of high importance in order to determine exactly which one, metabolites and/or minerals, are responsible for the described effects.

### 3.6. Other Animals Feeds

There has been research with the purpose of including seaweeds in the diet of other species. It is also worth mentioning the research made with rabbit feed. Mainly, there are some encouraging results with the inclusion of red and green seaweed species in the feed.

Low amounts of green seaweed, mainly from the *Ulva* genus, also showed encouraging results. A meal with 1% *Ulva* showed positive effects on growth performance and diet digestibility, at the same time as no hematological or biochemical parameters show negative effects on rabbit health [136]. The inclusion of higher than 5% rates is usually associated with no statistical differences between control groups and seaweed feed groups [137–140]. The potential of seaweed as rabbit feed requires more study to assess its full potential. The usage of calcified red seaweed, such as *Lithothamnium* sp. up to 1% mix could lead to the increase of calcium in rabbits, inferred by the observation of a reduction in width and length of the intestinal villi [9,141–143]. In both cases, there is a tendency to use lower amounts of seaweed, since the benefits disappear as the percentage is increased. The results also show potential research in bioavailability on mixing the two species in order to retain the best advantages of both in one prebiotic solution.

It is worth mentioning the use of seaweed, mainly brown algae in pig farms. Historically, it is described as the usage of a mixture of brown algae species (boiled or raw), like *Fucus vesiculosus*, *Pelvetia* sp. or *A. nodosum* with cereal meal to fatten pigs in Sweden and Scotland [144,145]. However, it was already proved that high amounts of brown seaweed can be detrimental to pigs, such as causing weight loss after several weeks feeding them with a 10% *A. nodosum* [146]. This kind of result led us to using seaweed as an addictive in low amounts (1–2%) for potential benefits in pigs' health and meat quality [9]. There are two main reasons to use seaweed on pig feed. One of them is the use of seaweed as a prebiotic and its health effects. The use of seaweed and seaweed extracts have been shown to have prebiotic effects and enhance immunologic function in pigs and have been assessed to replace antibiotics in pig farms [9]. There is proof that the use of polysaccharides as fucoidan and laminarin as an extract improves piglet performance, being that laminarin is the main source for gut health and performance improvements [147,148]. On the other hand, a few studies have been done with raw seaweed. There was a Japanese team trial, in which they fed pigs with 0.8% unspecified seaweed species feed for four days, from 76 day- to 80 day-old subjects, resulting in Immunoglobulin A production in saliva and immune function [149]. The work of Dierick et al. [150,151] tried to reproduce

their own in vitro results, which indicated that a 1% *A. nodosum* feed had a depressive effect on the gut flora, especially *E. coli*, while increasing the *Lactobacilli/E. coli* ratio and leading to resistance to intestinal disorders. However, seaweed meal added at 0.25, 0.5 and 1% to piglet diets, failed to enhance performance, gut health, plasma oxidative status and did not alter microbial ecology in the foregut and in the caecum [150,151]. It was latter theorized that this lack of effect may be due to phlorotannins in *A. nodosum*, which could counteract the prebiotic effects of other compounds due to a too low inclusion rate. Regarding the unchanged oxidative status, antioxidant vitamins in the diet may mask the antioxidant effect of seaweed [9,152]. Another main reason to use seaweed as pig feed is a proposed strategy to face the problem of lack of iodine in some populations. The usage of brown seaweed, such as *Laminaria* and *Ascophyllum*, to enrich pig's meat with organic iodine, which is readily metabolized and stored in pig muscle, is an easily controllable strategy, with no risk of overdosing, however limited, to achieve the referred propose [150,153]. Feed pigs with 2% of dried *A. nodosum* meal increased the concentration of iodine in the tissue by 2.7 to 6.8, depending on the tissue [150]. As said before, however limited, this strategy offers a solution by introducing iodine enriched food in human nutrition with a low risk.

There is some background of the use of seaweeds for diverse types of animals during human history, where the good results are mainly due to seaweed minerals and polysaccharides, enhancing the growth performance and potentiating digestibility of the normal diet, but the data are scarce, thus there is a need for more research in this area. One of the main topics which should be studied is related with the bioavailability of seaweed molecules during the livestock digestion. Such bioavailability studies are fundamental since it will allow not only to determine how much of the seaweed content is made available during the livestock digestion process but will also help to study the mechanisms in which some seaweed metabolites could counteract the effects of others in the same species or in a mixture of different seaweeds.

## 4. Conclusions

Seaweeds are close to becoming popular, due to their suitability as potential feedstock production, as well as supplements for food items. Seaweeds are rich in protein, dietary fibers and phytochemicals used to enhance the nutritional quality of animal feed. The increasing demand over renewable and sustainable energy sources without compromising on food and land resources can be fulfilled by seaweeds as they are fast growing, high biomass yielding with elevated and free of charge productivity, compared to other conventional biomass feedstock, such as corn or soybean.

The seaweed animal feed assays occur mainly as fresh seaweed, dried seaweed or even seaweed crude extract. There is a general lack of nutritional and biochemical studies of seaweed as feeds that makes the analysis of seaweed composition effect in the animal welfare difficult. Thus, more studies, regarding seaweed complete biochemical profile (macro and micronutrients, also seaweed metabolites), are needed to fully understand the impact of seaweeds in the animals.

However, seaweeds evidenced their potential to be further explored as an animal feed additive/supplement and cannot be applied as a complete substitute of the typical animal feed. Seaweed benefic effects are generally below 10% of the total concentration in the animal feed; above that, it was demonstrated to show negative effects and even animals refused to eat the provided feed.

Actually, with the active search for alternatives to the typical feed supplements and antibiotics, seaweed is one of the main hypotheses for animal feed supplementation, because seaweed production does not compete for arable land or fresh water. However, the wild seaweed biomass does not have a quality guarantee, because of the variations of nutritional values and risks of bioaccumulation of heavy metals, to provide a reliable source of safe animal feed supplementation. Consequently, seaweed aquaculture is the alternative solution for seaweed production and can be met through improvements in existing technology (already in use in Acadian Seaplants, seaweed Production Company from Canada).

**Author Contributions:** Conceived and designed the idea: T.M., A.I., T.C., M.M., J.C., K.B.; Organization of the team: J.C.; Writing and bibliographic research: T.M., A.I., T.C., M.M., J.C., K.B.; Supervision and Manuscript Revision, L.P. and K.B. All authors have read and agreed to the published version of the manuscript.

**Funding:** This work is financed by national funds through FCT—Foundation for Science and Technology, I.P., within the scope of the projects UIDB/04292/2020—MARE—Marine and Environmental Sciences Centre. João Cotas thanks the European Regional Development Fund through the Interreg Atlantic Area Program, under the project NASPA.

**Conflicts of Interest:** The authors declare no conflict of interest.

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
