# Peer review of "Seaweed Potential in the Animal Feed: A Review"

_jmse, doi:10.3390/jmse8080559_

Round 1
Reviewer 1 Report
I have carefully reviewed manuscript entitled: Seaweed Potential in the Animal Feed: a Review. This manuscript is very interesting and presents important data concerning the potential use of seaweeds in animal feeding.
The title is informative, clear and it is in good correlation with the article content. References are representative for the subject area. But I have some comments on this manuscript, which will help to improve it. In my opinion, manuscript should be accepted after minor revision.
Comments:
In my opinion English should be checked by Native Speaker, it must be improved
Line 39: I propose “Macroalgae, in general….”
Line 42: polyunsaturated fatty acids – please add abbreviation here (PUFA) – not Line 214. Abbreviations should appear where the full names are listed for the first time in the manuscript.
Line 47: “essential amino acids” – again please add here (EAAs) – not Line 135
Line 69: “There are known….”
Line 74: “count”
Line 75: “…are similar in length to red seaweeds”??? Please check
Line 78: “1% is known….”
Line 79 and 92 and 194: please keep the same form – “β-carotene” or “beta-carotene” or “β carotene”
Line 80: when you list polysaccharides typical to brown seaweeds, you should also add for green and red seaweeds
Line 83: “such as,….” – please delete double spaces in the manuscript
Line 85: I propose to prepare the same scheme for describing each group of algae, for example – first paragraph – minerals, then polysaccharides, proteins, lipids, etc. Now everything is mixed…
Line 95: “green seaweeds”
Line 99: I think it will be beneficial to add percentage chemical composition of Ulva, as an example, to have better imagination/comparison
Lines 107-116: this text is still about chemical composition so I propose to move it to Line 100
What about vitamins in green seaweeds? Please add
Line 121: this sentence is very general - brings no information. Please provide average values of iodine content in these seaweeds
Line 123: “Red algae possesses more cationic locations….” – not clear, please rewrite
Lines 121-129: here each paragraph concerns one sentence – please prepare separate paragraphs for each active compound
Lines 125-127: fat is mixed with fibers… “Didn’t find…”
Lines 128 and then 130: one paragraph
Line 131: “….aspartic acid, which compose….” Please provide here the abbreviation for the listed amino acids – not later as it is in Line 171
Line 135: delete EEAs
Line 139: add units (1.0-1.3%), (0.7-3%)
Line 140: 20:5, please remove space in the other fatty acids
Line 157: please add this level
Line 160: “In these algae can be found alginates and glucans [24].” – to the paragraph about polysaccharides
Line 167: “(Phaeophyceae)” – should be upper, in the Line 155
Line 173: lack of abbreviations for amino acids, unless previously stated
Line 177: please standardize in the whole manuscript n-3 or ω 3, or omega-3 (Line 257). Delete “Red” – this section is about brown seaweeds
Line 181: “C18:1ω9 (oleic acid)” – first oleic acid (ω-9, C18:1), add “-” in ω-9. The name should be analogous as in Line for example 179. Please correct later
Line 188: “Fucoidans, sulphated polysaccharides… .” concern polysaccharides, should be – e.g., Line 161
Line 188: it will be beneficial to add properties to all groups of active compounds, present in seaweed biomass, not only for polysaccharides
Line 197: This sentence should be earlier –first paragraph about minerals. What do you mean – “considerably higher”? Please add some examples to have any imagination about this amount.
In the case of seaweeds in animal feeding it is necessary to mention about heavy metals because they can disqualify algae for this use.
Line 200: provide examples, similar as for red seaweeds – Line 152.
Lines 200-215: this paragraph is about chemical composition of seaweeds, many information is repeated. It fits better to Line 67.
Line 212: “Can be found noticeable cases of iodine in brown and calcium in red seaweed” – what does it mean? Not clear
Line 214: please delete PUFAs – should be in Line 42
Line 223: “soybean, ….there are some problems related with the deficiency of some essential amino acids in such feed” and Line 228: “soybean show that this plant do not fully match the fish nutritional requirements, especially with respect to amino and fatty acids” – some repetitions, please avoid them
Line 232: I propose “..green Ulva and red seaweeds Gracilaria with the objective…” there is no use to repeat several times (Chlorophyta) and (Rhodophyta). It was mentioned at the beginning
Line 236: you can mention here about Integrated Multitrophic Aquaculture (IMTA)
Line 254: delete (Phaeophyceae), (Rhodophyta) – Line 428, (Phaeophyceae) – Line 434
Line 255: shouldn’t be: “e.g.” or “i.e.” instead of “p.e.”? – Line 291 – should be “for example”
Line 284: “…that grow…”. Please use abbreviation of the elements on the first page – Line 32.
Line 285: “sp.” without Italics
Line 298: “bio accessibility” – Line 305 “bio-accessibility”, please standardize in the whole manuscript
Line 300: “species”
Lines 301-304: this sentence should be rewritten, is incomprehensible
Lines 333-334: “…towards….U. rigida.” should be rewritten, is incomprehensible. Remove space between 10 and %
Line 337: should be for example 0.7-1.7%, use only once %. Please correct in the whole manuscript. And additionally dot instead of comma
Line 348: “seaweed has been”
Line 353: “livestock”
Line 363: “algae” – is plural, so “algae contain…”
Line 388-389: please rewrite this sentence
Line 405: “…has the potential….”
Line 436: full names of seaweeds
Line 453: This section is very interesting!
Line 462: “seaweed” is singular so should be “has been…”
Line 502: many single sentences in this section, please group them into separate paragraphs
Line 566: please avoid references in the Conclusions section
References:
Small letters in the title of publication – e.g., Line 609, 614, 616, etc. please check in the whole section
Author Response
Comment 1: I have carefully reviewed manuscript entitled: Seaweed Potential in the Animal Feed: a Review. This manuscript is very interesting and presents important data concerning the potential use of seaweeds in animal feeding.
The title is informative, clear and it is in good correlation with the article content. References are representative for the subject area. But I have some comments on this manuscript, which will help to improve it. In my opinion, manuscript should be accepted after minor revision.
Line 453: This section is very interesting!
In my opinion English should be checked by Native Speaker, it must be improved.
Answer 1: Firstly, we would like to thank the reviewer for his/her words. They were very valuable in improving the overall quality of the manuscript. We did an extensive revision at the English level.
Comment 2: Grammar and text problems:
Line 39: I propose “Macroalgae, in general….”
Line 42: polyunsaturated fatty acids – please add abbreviation here (PUFA) – not Line 214. Abbreviations should appear where the full names are listed for the first time in the manuscript.
Line 47: “essential amino acids” – again please add here (EAAs) – not Line 135
Line 69: “There are known….”
Line 74: “count”
Line 75: “…are similar in length to red seaweeds”??? Please check
Line 78: “1% is known….”
Line 79 and 92 and 194: please keep the same form – “β-carotene” or “beta-carotene” or “β carotene”
Line 80: when you list polysaccharides typical to brown seaweeds, you should also add for green and red seaweeds
Line 83: “such as,….” – please delete double spaces in the manuscript
Line 85: I propose to prepare the same scheme for describing each group of algae, for example – first paragraph – minerals, then polysaccharides, proteins, lipids, etc. Now everything is mixed…
Line 95: “green seaweeds”
Line 99: I think it will be beneficial to add percentage chemical composition of Ulva, as an example, to have better imagination/comparison
Lines 107-116: this text is still about chemical composition so I propose to move it to Line 100
What about vitamins in green seaweeds? Please add
Line 121: this sentence is very general - brings no information. Please provide average values of iodine content in these seaweeds
Line 123: “Red algae possesses more cationic locations….” – not clear, please rewrite
Lines 121-129: here each paragraph concerns one sentence – please prepare separate paragraphs for each active compound
Lines 125-127: fat is mixed with fibers… “Didn’t find…”
Lines 128 and then 130: one paragraph
Line 131: “….aspartic acid, which compose….” Please provide here the abbreviation for the listed amino acids – not later as it is in Line 171
Line 135: delete EEAs
Line 139: add units (1.0-1.3%), (0.7-3%)
Line 140: 20:5, please remove space in the other fatty acids
Line 157: please add this level
Line 160: “In these algae can be found alginates and glucans [24].” – to the paragraph about polysaccharides
Line 167: “(Phaeophyceae)” – should be upper, in the Line 155
Line 173: lack of abbreviations for amino acids, unless previously stated
Line 177: please standardize in the whole manuscript n-3 or ω 3, or omega-3 (Line 257). Delete “Red” – this section is about brown seaweeds
Line 181: “C18:1ω9 (oleic acid)” – first oleic acid (ω-9, C18:1), add “-” in ω-9. The name should be analogous as in Line for example 179. Please correct later
Line 188: “Fucoidans, sulphated polysaccharides… .” concern polysaccharides, should be – e.g., Line 161
Line 188: it will be beneficial to add properties to all groups of active compounds, present in seaweed biomass, not only for polysaccharides
Line 197: This sentence should be earlier –first paragraph about minerals. What do you mean – “considerably higher”? Please add some examples to have any imagination about this amount.
Line 200: provide examples, similar as for red seaweeds – Line 152.
Lines 200-215: this paragraph is about chemical composition of seaweeds, many information is repeated. It fits better to Line 67.
Line 212: “Can be found noticeable cases of iodine in brown and calcium in red seaweed” – what does it mean? Not clear
Line 214: please delete PUFAs – should be in Line 42
Line 223: “soybean, ….there are some problems related with the deficiency of some essential amino acids in such feed” and Line 228: “soybean show that this plant do not fully match the fish nutritional requirements, especially with respect to amino and fatty acids” – some repetitions, please avoid them
Line 232: I propose “..green Ulva and red seaweeds Gracilaria with the objective…” there is no use to repeat several times (Chlorophyta) and (Rhodophyta). It was mentioned at the beginning
Line 236: you can mention here about Integrated Multitrophic Aquaculture (IMTA)
Line 254: delete (Phaeophyceae), (Rhodophyta) – Line 428, (Phaeophyceae) – Line 434
Line 255: shouldn’t be: “e.g.” or “i.e.” instead of “p.e.”? – Line 291 – should be “for example”
Line 284: “…that grow…”. Please use abbreviation of the elements on the first page – Line 32.
Line 285: “sp.” without Italics
Line 298: “bio accessibility” – Line 305 “bio-accessibility”, please standardize in the whole manuscript
Line 300: “species”
Lines 301-304: this sentence should be rewritten, is incomprehensible
Lines 333-334: “…towards….U. rigida.” should be rewritten, is incomprehensible. Remove space between 10 and %
Line 337: should be for example 0.7-1.7%, use only once %. Please correct in the whole manuscript. And additionally dot instead of comma
Line 348: “seaweed has been”
Line 353: “livestock”
Line 363: “algae” – is plural, so “algae contain…”
Line 388-389: please rewrite this sentence
Line 405: “…has the potential….”
Line 436: full names of seaweeds
Line 462: “seaweed” is singular so should be “has been…”
Line 502: many single sentences in this section, please group them into separate paragraphs
Line 566: please avoid references in the Conclusions section
Answer 2: We addressed the problem and all the text was trimmed and rewritten. All suggestions have been addressed.
Comment 3: In the case of seaweeds in animal feeding it is necessary to mention about heavy metals because they can disqualify algae for this use.
Answer 3: Thank you for your advice, we added a new section to discuss that matter, mainly about the feed safety and heavy metals problems.
Comment 4: References: Small letters in the title of publication – e.g., Line 609, 614, 616, etc. please check in the whole section
Answer 4: We have reviewed and corrected all the problematic references.

Reviewer 2 Report
Dear authors,
thanks for sending this manuscript for review. I can see you have taken the time to read, and familiarize yourself with the literature.
However, I don't think paper is ready for publication without major revisions. The 'problem' is that it isn't more than a summary of articles, it is not a review.
A review would require a clear objective, research questions and a methodology that is reproducable. What kind of questions do you want to 'ask to' the literature? is it not described.
I was enthusiastic when I read about safety - that could be a concept to explore further and take a step further than just summarize. But that is mentioned only in the beginning.
Author Response
Comment 1: Dear authors, thanks for sending this manuscript for review. I can see you have taken the time to read, and familiarize yourself with the literature. However, I don't think paper is ready for publication without major revisions. The 'problem' is that it isn't more than a summary of articles, it is not a review. A review would require a clear objective, research questions and a methodology that is reproducable. What kind of questions do you want to 'ask to' the literature? is it not described. I was enthusiastic when I read about safety - that could be a concept to explore further and take a step further than just summarize. But that is mentioned only in the beginning.
Answer 1: We would like to thank the reviewer’s words. We are pleased with your feedback. We added more information and content to the manuscript and we hope tried to answer the question, mainly in the section 3.1 (Feed safety) and conclusion, where we focused the objective, obtained from the literature and the problems that we had during the analysis.

Reviewer 3 Report
This is a clear and well structured review on the potential use of seaweeds on animal feeds.
The strong points of the work are the clarity and the amount of information.
The weak points are the lack of originality (there is a number of other publications dealing on the same subject) and the absence of some tables and/or diagrams to resume and present information in a more attractive way (i.e. main features that distinguish or characterize the three types of seaweeds, main results obtained per type of animal, etc). Authors are encouraged to include these in order to get a more reader-friendly paper.
Author Response
Comment 1: This is a clear and well structured review on the potential use of seaweeds on animal feeds. The strong points of the work are the clarity and the amount of information. The weak points are the lack of originality (there is a number of other publications dealing on the same subject) and the absence of some tables and/or diagrams to resume and present information in a more attractive way (i.e. main features that distinguish or characterize the three types of seaweeds, main results obtained per type of animal, etc). Authors are encouraged to include these in order to get a more reader-friendly paper.
Answer 1: We thank the reviewer words. We added new tables in the section 2. We also added new information that we think can give new emphasis in the manuscript.

Round 2
Reviewer 2 Report
This paper summarizes previous studies in seaweed. Although some edits have been made in revising the manuscript, I still don't consider this a true review. It merely summarizes other publications.